# Considering layerwise importance in the Lottery Ticket Hypothesis

## Abstract

The recently-introduced Lottery Ticket Hypothesis (LTH) posits that it is possible to extract a sparse trainable subnetwork from a dense network using iterative magnitude pruning. By iteratively training the model, removing the connections with the lowest *global* weight magnitude and rewinding the remaining connections, sparse networks can be extracted. These sparse networks are referred to as *lottery tickets* and when fully trained, they reach a similar or better performance than their dense counterpart. Intuitively, this approach of comparing connection weights globally removes a lot of context about the relations between connection weights in their layer, as the weight distributions in layers throughout the network often differ significantly. In this paper, we study a number of different approaches that aim at recovering some of this layer distributional context by computing a connection importance value that is dependent on the weights of the other connections in the same layer. We then generalise the LTH to consider weight importance values rather than weight magnitudes.

Experiments using these importance metrics on several architectures and datasets, reveal interesting aspects on the structure and emergence of Lottery tickets. We find that given a repeatable training procedure, applying different importance metrics leads to distinct performant lottery tickets with little overlapping connections which strongly suggests that lottery tickets are not unique.

## 1 Introduction

The recent trend in machine learning to chase higher benchmark scores by adding additional parameters, has led to an explosive increase in the size of neural network architectures. A prime example of this phenomenon are the GPT models. While the first model in the family (Radford et al., 2018) has 117 million parameters, the latest model (Brown et al., 2020) already has a whopping 175 billion parameters which amounts to a $> 1000$ times increase. However, this explosive rise in parameters poses new problems.

Training a single transformer model with a parameter count of 213 millon — still orders of magnitude smaller than GPT-3 — using Neural Architecture Search emits as much CO2 as five cars during their lifetime (Strubell et al., 2020). Furthermore larger models typically need specialised hardware and a lot of computing power for training and inference, which constrain the ability of the model to run on mobile devices, thus limiting powerful models to well-funded institutions. Finally, research has shown that these large models are typically overparameterized and encode a lot of redundant information that can be removed (Denil et al., 2013).

To alleviate these issues, numerous approaches have been studied to scale down the number of parameters in a model, while still preserving (roughly) the same performance. This can be achieved by, e.g., designing parameter-efficient network structures (Sandler et al., 2018), or sparsifying existing neural network structures via pruning (le Cun, 1990; Hassibi & Stork, 1993; Han et al., 2015; Louizos et al., 2018; Molchanov et al., 2017).

Until recently, it was thought to be difficult to train sparse neural networks from scratch (Evci et al., 2019), which was further strengthened by the finding that over-parameterized network architectures are proven to lead to an optimal global minimum when training (Zou & Gu, 2019). As such the classical way to reduce parameter count was via the train-prune-finetune loop, in which a model is

first trained to completion, then redundant connections are pruned and finally the resulting network is finetuned

The recently introduced lottery ticket hypothesis (LTH) (Frankle & Carbin, 2019) challenged this notion and introduced a procedure to extract a sparse trainable network — a lottery ticket (LT) — from a dense network. This is done using a pruning criterion in the form of the global weight magnitude in combination with a gradual pruning procedure.

In this paper we study a refinement on this criterion by adding a notion of layerwise importance, which we introduce in section 2. We do this by considering a number of different weight rescaling methods, such that the comparisons are more calibrated between layers. Quantitative and qualitative comparisons between different importance measures and the baseline are reported in section 3. In addition, we shine light on the observable differences in the generated LTs (section 4), and determine how LTs emerge and differ when considering identical training conditions (section 5). A brief overview of related work is laid out in section 6 and finally, the paper is concluded in section 7.

The key observations of our study are that: i) given a fixed weight initialization, it is possible to extract different lottery tickets that have similar performance, but differ significantly in their structure, ii) these tickets have a noticeable amount of common connections which have low-variance across tickets, and iii) these stable common connections survive the LTH procedure even when the other weights in the model are reinitialized. Together these observations suggest that these connections might be a promising avenue towards finding LTs more efficiently.

## 2 THE LOTTERY TICKET HYPOTHESIS

The Lottery ticket hypothesis uses iterative Global Magnitude Pruning (GMP) (Han et al., 2015), which prunes individual connections that have the lowest weight magnitudes in a network. By repeating this process multiple times, it is possible to obtain a highly sparse network that when trained still reaches commensurate accuracy.

Later work by Frankle et al. (2020) introduced a modification to the LTH procedure by rewinding to parameters at iteration $t = k \ll m$, rather than resetting to the initial parameters at $t = 0$. By rewinding to a later iteration, the performance of the found lottery tickets was improved for complex networks at high sparsities. In the literature, this procedure is usually referred to as Lottery Ticket Rewinding (LTR) rather than the LTH. We will adopt this naming in the rest of the document.

While the exact mechanism behind the success of lottery tickets is not fully understood, one hypothesis, posited by Evci et al. (2022), is that due to the LTH/LTR procedure the resulting networks are already in the same loss basin as the fully trained dense network and as such the ticket can still converge to a performant solution during training. Pseudocode for the LTR in its initial form can be found in Algorithm 1.

---

**Algorithm 1** The LTR procedure

---

1: Initialize a model $M$ with parameters $\theta_0$
2: Pretrain $M$ for k iterations resulting parameters $\theta_{0,k}$
3: **for** $i \leftarrow 0, n$ **do**
4:     Train $M$ for m-k iterations, resulting in parameters $\theta_{i,m}$
5:     $\theta_{pooled} \leftarrow Pool(abs(\theta_{i,m}))$
6:     $p \leftarrow$ j-th percentile of $\theta_{pooled}$
7:     Prune all connections with $abs(\theta_{i,m}) < p$
8:     Rewind parameters of $M$ to $\theta_{0,k}$
9: **end for**

---

A weak aspect of globally pruning is that the only factor that determines whether a connection is pruned, is the magnitude of the connection weight. As such, connection weights from different layers are compared on a global scale, rather than within the layer. This disregards more complex factors such as the weight distribution within a layer and the number of remaining connections in the layer. In fact, due to the commonly used Kaiming Normal initialization (He et al., 2015), different layers are already initialized at different weight distributions as the standard deviation is inversely

|  | Initialised (0 it) | Pretrained (1000 it) | Fully Trained (78200 it) |
|---|---|---|---|
| Conv1 | [-0.47, 0.41] | [-0.57, 0.78] | [-1.07, 0.90] |
| Downsample1 | [-0.65, 0.66] | [-0.64, 0.63] | [-1.01, 1.11] |
| Conv17 | [-0.11, 0.10] | [-0.12, 0.10] | [-0.15, 0.19] |
| FC | [-0.30, 0.27] | [-0.29, 0.39] | [-0.45, 1.19] |

Table 1: Weight distributions [minimum and maximum] of selected layers in a ResNet-18 network at different steps in the training procedure.

proportional to the size of the layer, and as illustrated in Table 1, they also converge to different distributions. However, it should be noted that this difference in standard deviation actually already introduces a small implicit bias in the weights such that the weights are somewhat dependent on the size of the layer. Using importance metrics makes this bias both more explicit and stronger, dependent on which metric is used.

Disregarding these previously-mentioned factors might come at the cost of accuracy loss in the resulting ticket. In the worst case this can even lead to a phenomenon called 'layer-collapse' (Tanaka et al., 2020), a critical failure case of neural network pruning in which information can no longer flow from the input layer to the output layer of the network. This finds its origin in one or more intermediate layers being completely destroyed, thus rendering the resulting predictions invalid.

**Proposal.** Taking these issues into account, we propose a modification of the LTR procedure in the form of assigning an importance value to each connection representing its importance within the layer. This importance score is still dependent on the weight magnitude, but also considers the weight magnitudes of all other unpruned connections within the same layer. In doing so, it injects an additional notion of the layer distribution in the pruning process. Intuitively, this should alleviate the issue with layer-collapse and should improve the performance of the LTR procedure. Furthermore, it is possible to view this modification as a generalization of the LTR procedure, where the purely magnitude-based approach uses the identity mapping to calculate importance values. The only computational overhead incurred in our method is the calculation of importance scores, which is negligible. A pseudo code overview of the modified procedure can be found in Algorithm 2.

---

**Algorithm 2** The modified LTR procedure

---

1: Initialize a model $M$ with parameters $\theta_0$
2: Pretrain $M$ for k iterations resulting parameters $\theta_{0,k}$
3: **for** $i \leftarrow 1, n$ **do**
4:     Train $M$ for m-k iterations, resulting in parameters $\theta_{i,m}$
5:     $score_i \leftarrow$ layerwise-importance($abs(\theta_{i,m})$)
6:     $score_{pooled} \leftarrow Pool(score_i)$
7:     $p \leftarrow$ j-th percentile of $score_{pooled}$
8:     Prune all connections with $abs(score_i) < p$
9:     Rewind parameters to $\theta_{0,k}$
10: **end for**

---

In the same context of layerwise importance calculation, Lee et al. (2021) introduced a metric called the LAMP score, which follows a model-level distortion minimization perspective by calculating the $L_2$ norm of the connections w.r.t. the other unpruned connections in the layer. The authors also demonstrated that this results in more performant lottery tickets at a very high sparsity levels. Intuitively, adding a layerwise importance score would allow the LTR procedure to focus more on the layers with more connections — and more redundancy, and as such avoid removing connections from sparser layers, thus limiting the impact on model performance.

Additionally, as also noticed in Lee et al. (2021), we can view the use of importance scores in the LTR procedure as an heuristic to calculate layerwise pruning ratios in the context of local pruning rather global pruning.

## 2.1 Considered Importance Metrics

To calculate a layerwise importance value for a given connection at position $j$ in layer $i$, we need access to the weight matrix $\boldsymbol{W}_{i,:}$.

- $L_1$ normalization : $importance(i,j) = \dfrac{W_{i,j}}{\sum \boldsymbol{W}_{i,:}}$

- $L_2$ normalization : $importance(i,j) = \dfrac{W_{i,j}^2}{\sum \boldsymbol{W}_{i,:}^2}$ (Lee et al., 2021)

- Softmax : $importance(i,j) = \dfrac{exp(W_{i,j})}{\sum exp(\boldsymbol{W}_{i,:})}$

- Min-Max normalization : $importance(i,j) = \dfrac{W_{i,j} - min(\boldsymbol{W}_{i,:})}{max(\boldsymbol{W}_{i,:}) - min(\boldsymbol{W}_{i,:})}$

The goal of the considered importance measures is to make weight values from different layers comparable. To accomplish this, there are two main approaches.

The first approach, as followed by $L_1$, $L_2$ and SoftMax is the sum-to-one principle, to rescale the importances of a layer such that the total importance of a layer is one. Then the importances themselves are still determined by (an operation on) the magnitude, but rescaled.

The second approach, followed by Min-Max normalization determines the importances such that each layer has the same lower and upper boundaries.

**Importance Distributions.** In case of the softmax function, we notice that the importance distribution is very narrow and the lower bound is exactly $1/|W|$, while for the other functions the distribution is more broad with a lower bound of $0$. This is due to a combination of two factors: (1) the Kaiming Normal initialization which has a mean of 0 and a standard deviation that is inversely proportional to the size of the layer and (2) the exponential function for which $e^x \approx 1$ if $|x| \ll 1$. Together this nullifies the dependence on the weights themselves almost completely in the softmax function when applied on large layers. $L_2$ is another measure that transforms the relevance distribution to be more narrower than the weight distribution due to the quadratic function. The other measures, $L_1$ and Min-Max, do not fundamentally impact the relevance distribution, only re-scale (and re-center) the distribution.

## 3 Experiments

Our experiments, are based on the `openlth` library (Frankle, 2020). We follow the configurations specified in Frankle et al. (2021) unless otherwise stated.

**Networks.** We adopt three different types of Neural Network architectures: Fully Connected Networks (LeNet-300-100 Lecun et al. (1998)), classical Convolutional Networks (VGG-16 Simonyan & Zisserman (2015)) and Residual Networks He et al. (2016) (ResNet-18 & ResNet-20).

**Datasets.** We adopt three different datasets commonly used in the LTH/LTR literature: MNIST, CIFAR-10 and TinyImageNet. MNIST is used in conjunction with LeNet-300-100, CIFAR-10 is used with both VGG-16 and ResNet-20, and TinyImageNet is used with ResNet-18.

For a detailed description of the used training configuration, we refer the reader to Appendix A. We consider the Lottery Ticket Hypothesis with weight rewinding (LTR) introduced in Frankle et al. (2020). The authors noted that inclusion of rewinding increased the performance of the procedure on all types of models and datasets, but more specifically on complexer datasets and models.

## 3.1 Quantitative evaluation

The goal of the initial experiments is to determine whether using different importance methods results in Lottery tickets with similar or better accuracy. To this end, we calculate the Top-1 accuracy results for Lottery Tickets at different sparsity levels. Results are listed in tabular form in Table 2 (averaged over 3 runs), or in graphical form in Appendix B.1. To enable comparison with other

Table 2: Average top-1 accuracy of different network & dataset combinations over 3 runs. **Boldfaced** results highlight the best result for a given amount of pruning steps.

(a) LeNet-300-100 on MNIST

| Steps (*sparsity*) | Magnitude | $L_1$ | $L_2$ | SoftMax | MinMax |
|---|---|---|---|---|---|
| Dense network | $98.11\% \pm 0.08$ | $98.11\% \pm 0.08$ | $98.11\% \pm 0.08$ | $98.11\% \pm 0.08$ | $98.11\% \pm 0.08$ |
| 5 (*67.23%*) | $98.00\% \pm 0.09$ | $\mathbf{98.12\% \pm 0.10}$ | $98.04\% \pm 0.15$ | $98.00\% \pm 0.08$ | $98.02\% \pm 0.09$ |
| 10 (*89.26%*) | $98.08\% \pm 0.06$ | $\mathbf{98.15\% \pm 0.05}$ | $98.06\% \pm 0.08$ | $98.05\% \pm 0.08$ | $98.04\% \pm 0.10$ |
| 15 (*96.48%*) | $97.94\% \pm 0.08$ | $\mathbf{98.12\% \pm 0.15}$ | $97.95\% \pm 0.10$ | $97.90\% \pm 0.05$ | $97.77\% \pm 0.17$ |
| 20 (*98.85%*) | $97.23\% \pm 0.15$ | $97.44\% \pm 0.02$ | $\mathbf{97.47\% \pm 0.10}$ | $97.33\% \pm 0.17$ | $97.41\% \pm 0.05$ |
| 24 (*99.53%*) | $90.62\% \pm 0.29$ | $90.89\% \pm 0.22$ | $91.05\% \pm 1.09$ | $91.29\% \pm 0.79$ | $\mathbf{91.82\% \pm 0.16}$ |

(b) ResNet-20 on CIFAR-10

| Steps (*sparsity*) | Magnitude | $L_1$ | $L_2$ | SoftMax | MinMax |
|---|---|---|---|---|---|
| Dense network | $91.67\% \pm 0.40$ | $91.67\% \pm 0.40$ | $91.67\% \pm 0.40$ | $91.67\% \pm 0.40$ | $91.67\% \pm 0.40$ |
| 5 (*67.23%*) | $\mathbf{92.01\% \pm 0.14}$ | $91.57\% \pm 0.20$ | $91.65\% \pm 0.25$ | $90.15\% \pm 0.92$ | $91.64\% \pm 0.13$ |
| 10 (*89.26%*) | $\mathbf{90.90\% \pm 0.15}$ | $90.56\% \pm 0.22$ | $90.86\% \pm 0.39$ | $87.75\% \pm 0.33$ | $90.51\% \pm 0.05$ |
| 15 (*96.48%*) | $85.59\% \pm 1.00$ | $84.76\% \pm 0.62$ | $\mathbf{86.28\% \pm 0.45}$ | $84.00\% \pm 0.54$ | $83.28\% \pm 0.32$ |
| 20 (*98.85%*) | $\mathbf{77.07\% \pm 1.57}$ | $76.31\% \pm 0.71$ | $76.68\% \pm 0.63$ | $76.37\% \pm 1.06$ | $74.36\% \pm 0.57$ |
| 25 (*99.62%*) | $63.36\% \pm 0.55$ | $\mathbf{63.40\% \pm 0.77}$ | $62.78\% \pm 0.46$ | $63.07\% \pm 0.55$ | $50.17\% \pm 6.78$ |
| 30 (*99.88%*) | $\mathbf{45.66\% \pm 2.96}$ | $45.36\% \pm 1.49$ | $43.65\% \pm 0.80$ | $38.42\% \pm 5.46$ | $31.50\% \pm 1.80$ |
| 35 (*99.96%*) | $\mathbf{22.58\% \pm 4.57}$ | $21.60\% \pm 4.79$ | $19.59\% \pm 6.37$ | $17.75\% \pm 3.20$ | $14.62\% \pm 4.00$ |
| 40 (*99.99%*) | $12.87\% \pm 4.97$ | $10.00\% \pm 0.00$ | $12.45\% \pm 3.55$ | $12.29\% \pm 2.00$ | $\mathbf{13.34\% \pm 2.89}$ |

(c) ResNet-18 on TinyImageNet

| Steps (*sparsity*) | Magnitude | $L_1$ | $L_2$ | SoftMax | MinMax |
|---|---|---|---|---|---|
| Dense network | $49.53\% \pm 0.56$ | $49.53\% \pm 0.56$ | $49.53\% \pm 0.56$ | $49.53\% \pm 0.56$ | $49.53\% \pm 0.56$ |
| 5 (*67.23%*) | $49.99\% \pm 0.10$ | $\mathbf{50.96\% \pm 0.19}$ | $50.45\% \pm 0.40$ | $\mathbf{50.96\% \pm 0.18}$ | $50.38\% \pm 0.47$ |
| 10 (*89.26%*) | $49.98\% \pm 0.41$ | $\mathbf{50.84\% \pm 0.65}$ | $50.49\% \pm 0.50$ | $50.69\% \pm 0.73$ | $50.24\% \pm 0.68$ |
| 15 (*96.48%*) | $48.72\% \pm 0.20$ | $\mathbf{49.76\% \pm 0.56}$ | $49.41\% \pm 0.09$ | $45.30\% \pm 4.42$ | $47.77\% \pm 0.37$ |
| 20 (*98.85%*) | $46.36\% \pm 0.52$ | $\mathbf{47.58\% \pm 0.24}$ | $47.26\% \pm 0.52$ | $40.48\% \pm 3.52$ | $43.49\% \pm 0.17$ |
| 25 (*99.62%*) | $\mathbf{38.54\% \pm 0.48}$ | $37.37\% \pm 0.35$ | $37.90\% \pm 0.54$ | $32.51\% \pm 2.57$ | $34.23\% \pm 1.19$ |
| 30 (*99.88%*) | $24.56\% \pm 1.00$ | $25.12\% \pm 0.71$ | $\mathbf{25.88\% \pm 0.47}$ | $21.67\% \pm 1.40$ | $23.34\% \pm 0.12$ |
| 35 (*99.96%*) | $13.56\% \pm 0.11$ | $14.03\% \pm 0.61$ | $\mathbf{15.30\% \pm 0.28}$ | $10.16\% \pm 2.17$ | $12.74\% \pm 0.76$ |
| 40 (*99.99%*) | $4.62\% \pm 0.25$ | $6.48\% \pm 0.30$ | $\mathbf{6.83\% \pm 0.36}$ | $4.69\% \pm 0.72$ | $5.31\% \pm 0.62$ |

(d) VGG16 on CIFAR-10

| Steps (*sparsity*) | Magnitude | $L_1$ | $L_2$ | SoftMax | MinMax |
|---|---|---|---|---|---|
| Dense network | $93.52\% \pm 0.08$ | $93.52\% \pm 0.08$ | $93.52\% \pm 0.08$ | $93.52\% \pm 0.08$ | $93.52\% \pm 0.08$ |
| 5 (*67.23%*) | $93.45\% \pm 0.12$ | $93.57\% \pm 0.14$ | $93.69\% \pm 0.04$ | $93.49\% \pm 0.03$ | $\mathbf{93.76\% \pm 0.25}$ |
| 10 (*89.26%*) | $93.75\% \pm 0.03$ | $\mathbf{93.78\% \pm 0.25}$ | $93.57\% \pm 0.06$ | $64.99\% \pm 47.64$ | $93.67\% \pm 0.11$ |
| 15 (*96.48%*) | $93.69\% \pm 0.10$ | $\mathbf{93.76\% \pm 0.04}$ | $93.66\% \pm 0.18$ | $64.68\% \pm 47.36$ | $93.47\% \pm 0.14$ |
| 20 (*98.85%*) | $\mathbf{93.53\% \pm 0.10}$ | $93.24\% \pm 0.14$ | $93.03\% \pm 0.04$ | $63.72\% \pm 46.54$ | $92.83\% \pm 0.02$ |
| 25 (*99.62%*) | $91.88\% \pm 0.14$ | $90.94\% \pm 0.31$ | $\mathbf{91.48\% \pm 0.30}$ | $60.66\% \pm 43.87$ | $91.81\% \pm 0.09$ |
| 30 (*99.88%*) | $51.53\% \pm 36.01$ | $83.70\% \pm 0.15$ | $\mathbf{84.03\% \pm 0.31}$ | $56.58\% \pm 40.35$ | $33.95\% \pm 22.7$ |
| 35 (*99.96%*) | $10.00\% \pm 0.00$ | $\mathbf{40.99\% \pm 1.12}$ | $29.54\% \pm 6.67$ | $24.21\% \pm 17.48$ | $14.38\% \pm 7.59$ |

studies, we adopted the commonly used pruning ratio of 20%. Additionally, we will make our code public at `github-link-here` to foster reproducibility of the reported results.

From these experiments, we can distill a few key observations on which we elaborate in the following sections. First, the magnitude, $L_1$ and $L_2$ criterions all produce LTs of similar performance, the only noticeable difference is in the VGG16 + CIFAR10 experiment. Second, Min-Max normalization and SoftMax seem to keep up with magnitude pruning during the initial pruning phase, but suffers during the later iterations. This seem to be the trend except in the simplest scenario.

## 3.2 SIMILARITY BETWEEN $L_1$, $L_2$ AND MAGNITUDE

We notice that the top-1 accuracy of both magnitude, and the $L_1$ and $L_2$ normalization are pretty much identical at the earlier pruning ratios, contradicting our initial intuition. Our hypothesis is that this lack of difference indicates that the approaches to finding lottery tickets are less strict than initially thought, as introducing the notion of importance does not seem to have a large impact on the LTH procedure. We elaborate further on this more in the rest of the paper, where we conduct a deeper study on some differences between the found lottery tickets.

## 3.3 ISSUES WITH MIN-MAX

Due to the definition of Min-Max normalization, at least one connection in each layer will have an importance of 0 and at least one connection in each layer has an importance of 1. This will mean that

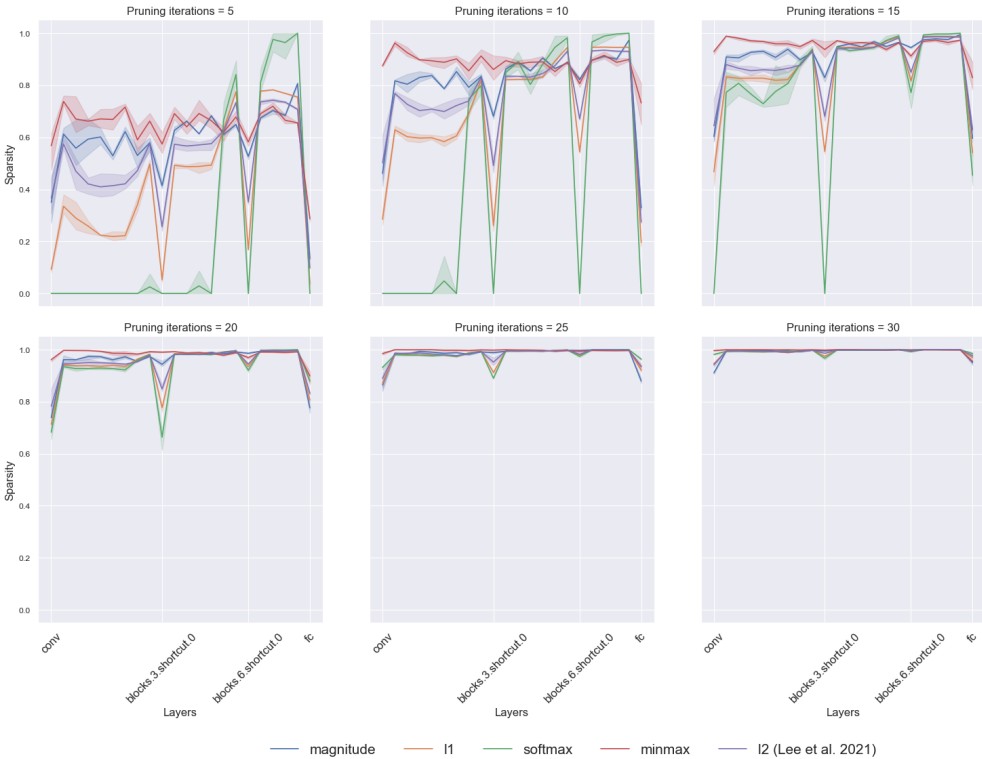

Figure 1: How the layer sparsities evolve during the LTH procedure with the ResNet-18 model trained on TinyImageNet. Layers of interest are annotated on the X-axis.

during each pruning iteration, connections will be pruned from each layer. Eventually, this means that the the thinner layers will have too much capacity removed which will affect the final accuracy. This phenomenon can also be seen in the additional figures of Appendix C.

### 3.4 ON LAYER-COLLAPSE

One of the possible symptoms of Layer-collapse is that the predictions of the model revert to random chance. Indeed, we notice in the later iterations of the LTH procedure on VGG16 with CIFAR-10, that the top-1 accuracies of the magnitude criterion as well as one run of the softmax criterion are equivalent to random chance.

Additionally, Layer-collapse also occurs in the ResNet experiments, albeit less noticeable. More specifically this occurs with the SoftMax and Magnitude criterions, but due to the presence of skip connections the flow of information is still preserved when the main connection is fully pruned. As such, the resulting top-1 accuracies are not nearly as affected.

## 4 A CLOSER LOOK

In this section we take a closer look at the generated LTs from the experiments. We focus our study on how they differ in structure and how they evolve throughout the pruning process.

Taking a look at how the sparsity of different network layers evolves at different intervals in the LTH procedure (see Figure 1 for the plots from ResNet18 trained on TinyImageNet), we can clearly see that during the initial pruning iterations there are large differences both between different layers and between importance measures. Even though these large differences exist between the tickets generated by different importance measures, we established in the previous section that there is no significant difference in the top-1 accuracy of Magnitude, $L_1$ and $L_2$. This is an indication that different kinds of structures exist in a network, that when trained can still achieve a competitive

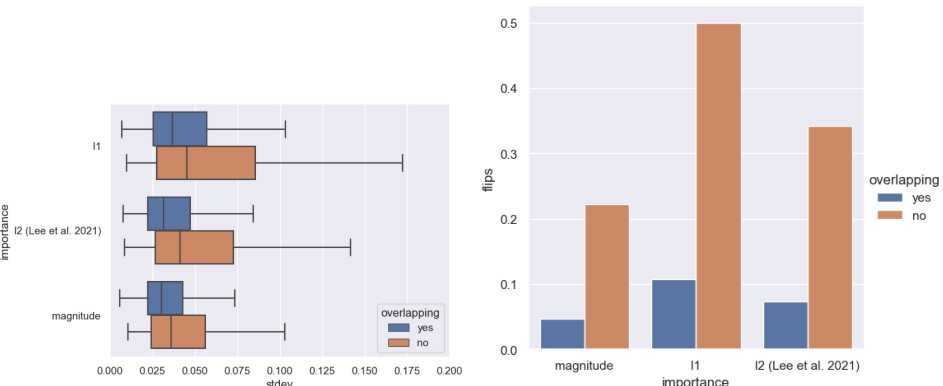

Figure 2: **Left:** The distribution of standard deviations, **Right:** The number of sign flips for overlapping and non-overlapping connections at 96.48% sparsity for the selected importance measures.

accuracy. As such, we might conclude that there are multiple ways to find Lottery Tickets (on which we expand in section 5). In later iterations, the difference between structures is still noticeable, but much less pronounced, which is expected as the number of pruned connections vastly outnumbers the number of unpruned connections.

Additionally, we can find an interesting pattern in the generated structures. We find that in each case, certain layers are pruned (significantly) less than most other layers in the network. Specifically, these layers are often layers that intuitively are more important to the quality of the training procedure. In the case of ResNet-18, these layers are both the input and output layer of the network, as well as the $1 \times 1$ convolutions used in the skip connections.

## 5 MULTIPLE LTS PER INITIALIZATION

In this experiment we determine whether a single initialization may lead to multiple Lottery Tickets. This requires making the LTR procedure deterministic and repeatable, i.e., all randomness in the training procedure is dependent on a set random seed. This ensures that the only impact on the quality and structure of the found ticket is the importance measure applied for the pruning step.

We consider the LTs found by the following importance measures: *magnitude*, $L_1$ and $L_2$. We limit ourselves to these measures, as they demonstrate a matching accuracy with the orginal dense network on the TinyImageNet dataset at a higher sparsity level (96.48%) than the other measures. As such, we also do this experiment on the same configuration namely ResNet18 and TinyImagenet.

We find that the resulting tickets share little amount of connections. Only 0.33% of the total (pruned and unpruned) connections are overlapping, while 9.34% of the unpruned connections are overlapping between different settings. We also find that some layers have a noticeably larger fraction of overlapping connections, more specifically the first few convolutional layers as well as the linear classification layer. This can be attributed to the intuition that these layers have less connections and as such are much less overparameterized.

Looking at the overlapping connections in the first convolutional layer (Figure 3), which is 68.78% of the remaining connections, compared to the non-overlapping connections, we can notice that the overlapping connections on average have a larger difference between the initial and final weight value. Please see the appendix for an extended set of plots. We also study whether these overlapping connections converge to the same weight in each pruning step up until the winning ticket is found. To do this, we have two simple metrics. For the first metric, we take the set of all weight values a single connection has at each pruning iteration and consider the standard deviation of that set as an indication of the robustness of that connection. We repeat this process for all connections in the ticket and find that on average the overlapping connections have a lower standard deviation than the non-overlapping connections. In Figure 2 we present results for a sparsity level of 96.48%. Moreover, it is worth noting that these trends are consistent over other sparsity levels as well (see subsection B.2).

The second metric we also apply is counting the number of sign flips. Once again considering the set of weight values a connection has during the pruning steps, we define a sign flip, if the sign of at least one instance in the set is differing from the others. Here too, we find that the overlapping connections have less sign flips than the non-overlapping connections. These results are presented in Figure 2, with results for other sparsity levels reported in the appendix.

As a final sanity check, we do a *partial reinitialization* test, where rather than starting the LTH/LTR procedure from the initial parameters $\theta_0$, we preserve the weights of $\theta_0$ that correspond to the overlapping connections at the 96.84% sparsity level, while reinitializing the weights of the remaining connections using the same distribution (Kaiming Normal). We then restart the LTR procedure with the studied importance measures using this initialization with the same deterministic training procedure as the initial initialization. The goal is to determine whether these overlapping connections are still (mostly) unpruned given a different initialization, and determine as such whether the overlapping property is dependent on the total initialization, or whether these connections are capable by themselves to steer an initialization in a similar, performant lottery ticket.

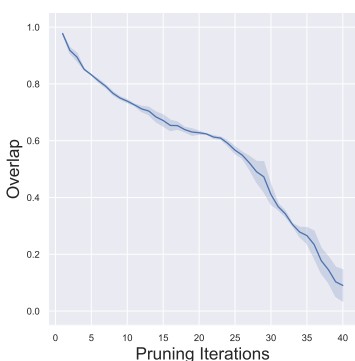

Figure 3: The evolution of overlapping connections within the first convolutional layer

We find that on average, for three random runs, the amount of remaining connections is higher than the expected value, assuming that each connection has the same chance to be in the ticket. More specifically, for the initial few convolutional layers, we see increases over the expected amount ranging from +8.5% to +70.0%. In the fully connected layer, we can see even higher increases ranging from +440% to +480%. Finally, these effects were not nearly as noticeable in the $1\times1$ convolutions, with only -3.2% to +20.0%.

These results suggest that these connections are, by grace of their individual initializations, more likely to be part of the ticket. This also indicates that there likely is a minimum amount of connections in some layers that are necessary for a performant ticket, with those mostly concentrated in the initial layers and the classification layer.

## 6 RELATED WORK

### 6.1 PRUNING NEURAL NETWORKS

Neural networks can be trimmed of their fat by neural network pruning. These pruning methods can be grouped on a number of different axes : iterative pruning vs single-shot pruning, local pruning vs global pruning, data-driven vs data-free and more. Here, we limit ourselves and give a brief overview of different method categories that have some relation with respect to the analysis conducted in this paper. For an in-depth review, we refer the reader to Hoefler et al. (2021).

**Criterion-based.** This group constitutes the original line of research, which follows the train-prune-finetune approach, where the network is first trained on the dataset, then pruned using a criterion and finally the pruned network is finetuned on the task to regain some accuracy. Pioneered by le Cun (1990); Hassibi & Stork (1993), there has recently been a resurgence in popularity (Han et al., 2015; Li et al., 2017; Zhao et al., 2019; Dong et al., 2017; Yu et al., 2018).

**Foresight Pruning.** Inspired by the LTH, this line of research tries to prune a network at initialization by using a small subset of the training data to select promising connections, after which the sparse network will be fully trained. (Wang et al., 2020; Lee et al., 2019; Wang et al., 2019; Tanaka et al., 2020) Recent work (Liu et al., 2022) asserted that these methods are functionally distinct to the LTH, because rather than generating sparse structures that depend on the initial weights, these methods learn efficient layerwise sparsity ratios.

**Pruning and Regrowing.** This line of research has been inspired by biological processes in which new pathways in the brain grow during the lifetime and involves methods that learn both weights and connections during the training process. (Guo et al., 2016; Bellec et al., 2018; Mocanu et al., 2018; Dai et al., 2019)

**Regularization-based.** Uses a regularization term on the weights of network during training that will be minimized, such that a sparsity is automatically learned during the training process. (Huang & Wang, 2018; Alvarez & Salzmann, 2016; Srinivas et al., 2017; Liu et al., 2017)

## 6.2 THE LOTTERY TICKET HYPOTHESIS

Introduced by Frankle & Carbin (2019), the Lottery Ticket Hypothesis introduces a method to extract sparse trainable networks from a dense network. Initially, this method was criticised due to the reliance on low learning rates (Liu et al., 2019) and the fact that it could not be reproduced in more complex settings (Gale et al., 2019). However, these concerns were mitigated with the introduction of weight rewinding (Frankle et al., 2020). These initial papers sparked a large body of follow-up work that we will briefly summarize in the next paragraphs.

**Empirical and Theoretical properties.** A first line of follow-up work was focussed on the properties of the LTH. Zhou et al. (2019) systematically explored different components of the LTH procedure. Renda et al. (2020) compares the impact of finetuning and rewinding and introduces the aspect of learning rate rewinding. Maene et al. (2021) introduces a theoretical proof for the effectiveness of the LTH, while Evci et al. (2022) studied the gradient flow in the LTH procedure. Given the characteristics of our study, our work is situated in this category.

**Transfering LTs.** A second line of research involved transfering lottery tickets. This includes the transferability of lottery tickets between different datasets (Morcos et al., 2019; Soelen & Sheppard, 2019; Mehta, 2019; Sabatelli. et al., 2021; Desai et al., 2021; Chen et al., 2021a) and even between different models (Chen et al., 2021d).

**Other domains and models.** A third line of research studied the application of the LTH approach — which was initially introduced for Convolutional Networks — on other types of models and domains. (Chen et al., 2021c; Diffenderfer & Kailkhura, 2021; Chen et al., 2020; Kalibhat et al., 2021; Chen et al., 2021b)

**Strong LTH.** Finally, recently a lot of effort has been poured in the so-called 'strong Lottery Ticket Hypothesis' (Ramanujan et al., 2020) and proving its existence first in MLPs, then in CNNs and even in Equivariant Networks. (Chijiwa et al., 2021; da Cunha et al., 2022; Malach et al., 2020; Orseau et al., 2020; Pensia et al., 2020; Ferbach et al., 2022) The strong Lottery Ticket Hypothesis dictates that a fully trained network can be approximated by pruning a sufficiently large randomly-initialized network, rather than extracting a sparse network from a trained network as in the 'original' Lottery Ticket Hypothesis formulation.

## 7 CONCLUSION

We studied a modification to the Lottery Ticket Hypothesis by introducing the notion of layerwise importance in the procedure. While we did not note a significant difference in performance when using certain importance measures, we emphasize that the LTH is invariant in a certain sense to the layerwise pruning ratios, as long as the lowest magnitude weights are removed from each layer.

Furthermore, we noticed that given a fixed initialization, it is possible to extract different lottery tickets that are dependent on the importance measure that is used. These Lottery Tickets all have roughly the same top-1 accuracy, but differ significantly in their structure. Looking at the remaining connections in the tickets, we noticed that there exist a noticeable amount of overlapping connections between the tickets, namely mostly in the first convolutional layers and in the output layer. We observed that these overlapping connections consistently have a lower variance between different lottery tickets, suggesting that these connections are somehow more stable and converge more often to roughly the same value. Finally, we have shown with a partial reinitialization test, that these connections even survive when the other weights are reinitialized. This suggests that we should focus our attention on these specific connections to find LTs more efficiently.

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

# Supplementary Material

# Supplementary Material

## A    NETWORKS, DATASETS AND TRAINING

| Network | Dataset | Epochs | Batch | Opt. | Mom. | LR | LR Drop | Weight Decay | Initialization | Iters per Ep | Rewind Iter |
|---------|---------|--------|-------|------|------|-----|---------|--------------|----------------|--------------|-------------|
| LeNet | MNIST | 40 | 128 | SGD | — | 0.1 | — | — | Kaiming Normal | 469 | 0 |
| ResNet20 | CIFAR-10 | 160 | 238 | SGD | 0.9 | 0.1 | 10x at epochs 80, 120 | 1e-4 | Kaiming Normal | 391 | 1000 |
| VGG16 | CIFAR-10 | 160 | 128 | SGD | 0.9 | 0.1 | 10x at epochs 80, 120 | 1e-4 | Kaiming Normal | 391 | 2000 |
| ResNet18 | TinyImageNet | 200 | 256 | SGD | 0.9 | 0.2 | 10x at epochs 100, 150 | 1e-4 | Kaiming Normal | 391 | 1000 |

## B    ADDITIONAL FIGURES

### B.1    ACCURACY CURVES

The curves in Figure 4 illustrate the evolution of the top-1 accuracy of Lottery Tickets at different sparsities. These figures provide a more general overview than the tables the tables found in section 3, but are harder to extract fine-grained details from.

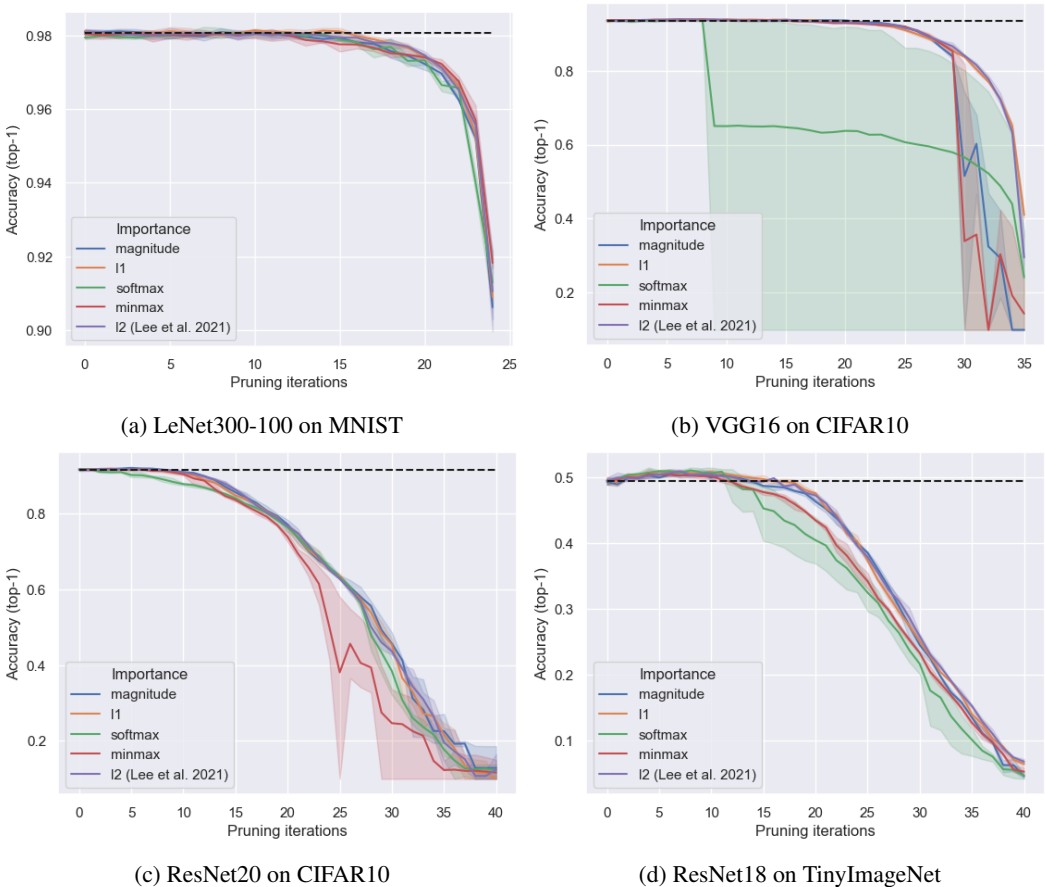

(a) LeNet300-100 on MNIST

(b) VGG16 on CIFAR10

(c) ResNet20 on CIFAR10

(d) ResNet18 on TinyImageNet

Figure 4: The accuracy curves for the different training configurations used in the main paper. The mean and standard deviation are given for 3 random seeds. The dashed line represents the mean accuracy of the dense network.

## B.2 MULTIPLE LTS PER INITIALIZATION

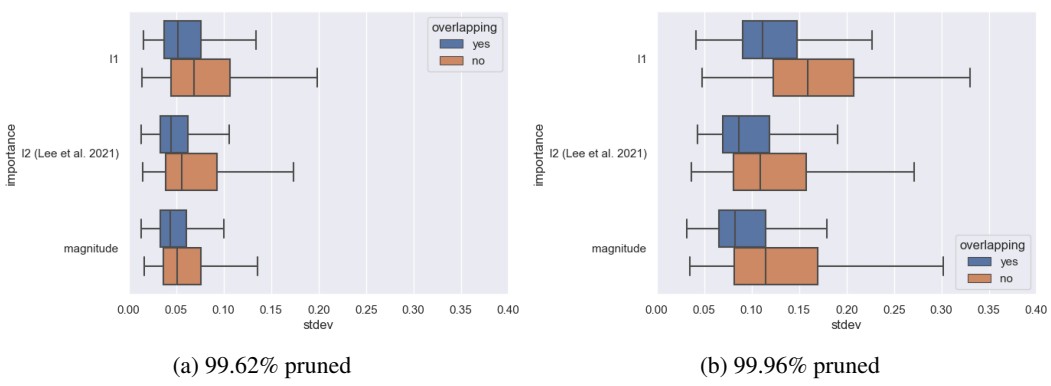

(a) 99.62% pruned

(b) 99.96% pruned

Figure 5: Additional distributions of standard deviations of both overlapping and non-overlapping connections for the {L1, L2, Magnitude} importance measures

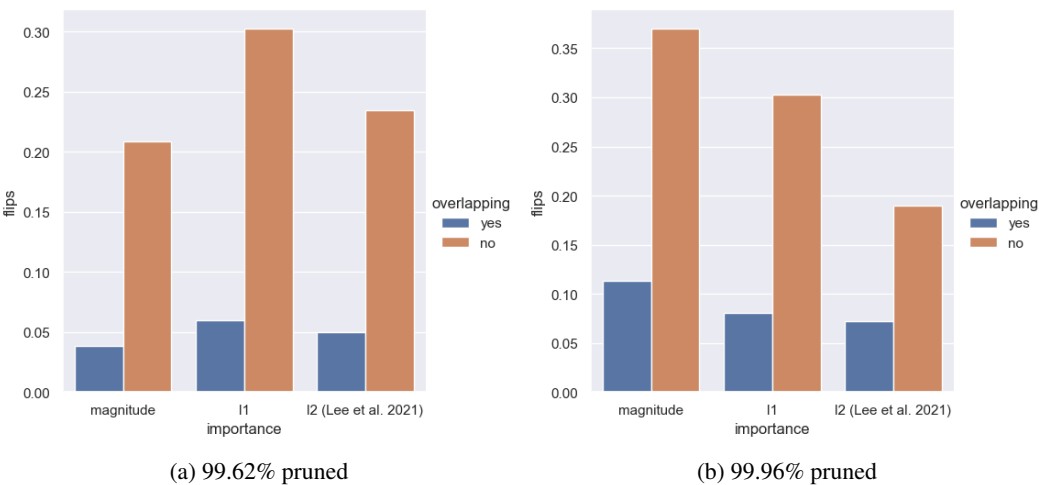

(a) 99.62% pruned

(b) 99.96% pruned

Figure 6: Additional Sign flips of both overlapping and non-overlapping connections for the {L1, L2, Magnitude} importance measures

## B.3   OVERLAP CURVES

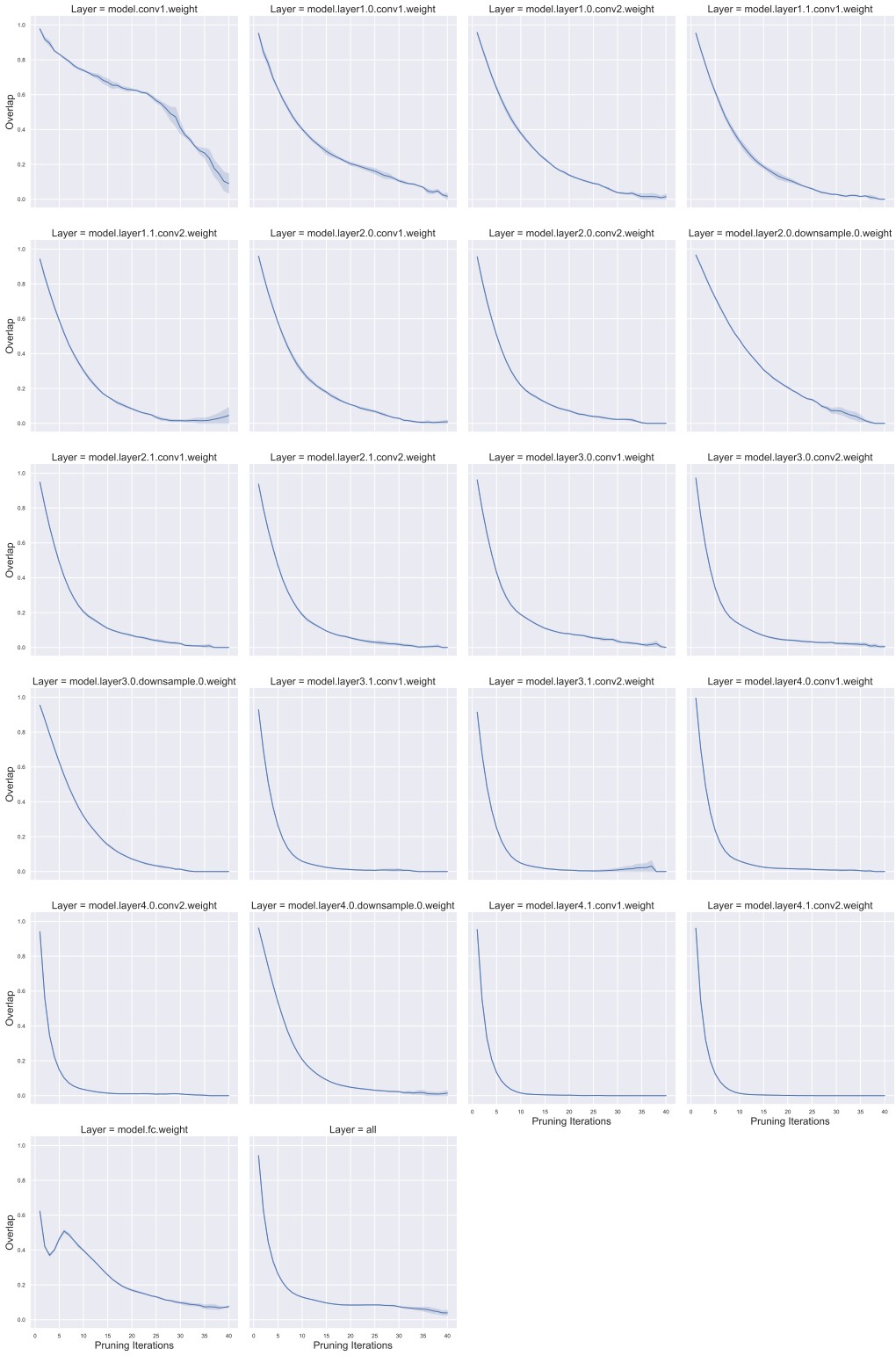

Figure 7: Additional curves measuring the proportion of overlapping connections between tickets

# C   LAYER SPARSITY

When studying the layerwise sparsities generated by the different pruning criteria (see Figure 8), we can notice a number of interesting observations. Primarily, we notice that for each pruning criterion, both the input and output layer are pruned proportionally much less than other layers, even though this wasn't explicitly encoded in the criterion. While the exact mechanism behind this isn't fully understood and is out-of-scope, we can theorize that this is due a combination of the training process and the fact that these layers are vital for the flow of information in the network. From all considered importance measures, the magnitude measure prunes the least of those layers.

When considering ResNets, we can additionally notice that most of the other criterions ($L_1$, $L_2$, SoftMax) also prune the 1x1 convolutions less. These 1x1 convolutions are located in the bottleneck modules and often contain less redundancy than the other 3x3 convolutions.

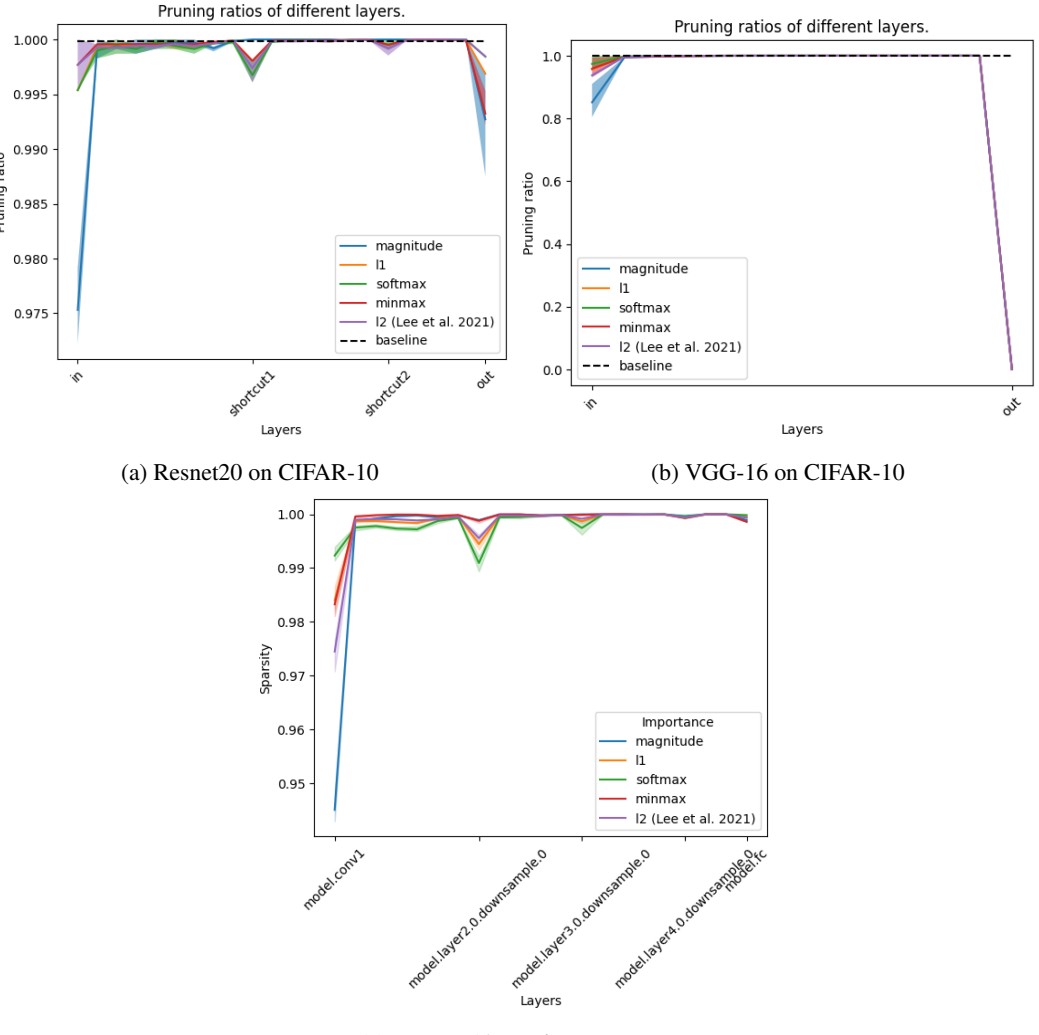

(a) Resnet20 on CIFAR-10    (b) VGG-16 on CIFAR-10

(c) ResNet-18 on TinyImageNet

Figure 8: The layerwise pruning ratio of different layers in a LT compared to the total pruning ratio. The x-axis is (roughly) corresponding to model depth and layers of interest are indicated.

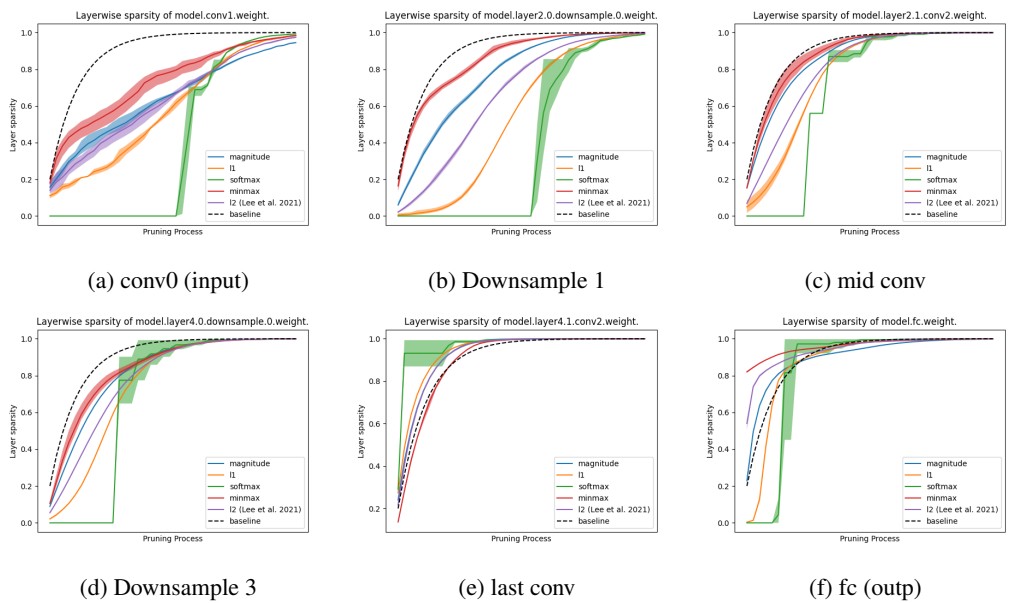

Figure 10: The evolution of layer sparsity during the LT extracting process, visualised in a number of selected layers of the ResNet18 network.

# D  LAYER SPARSITY IN TIME

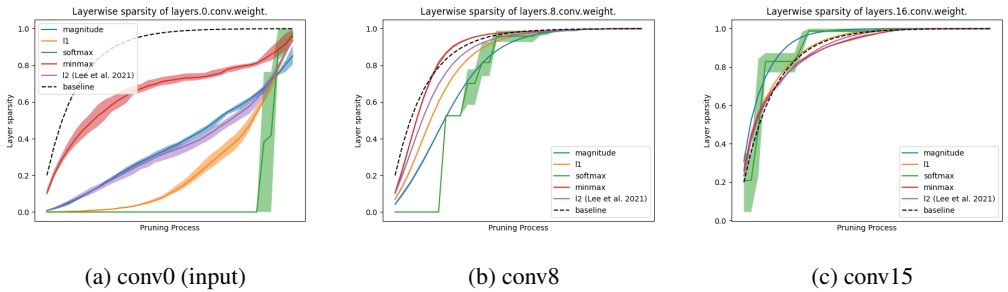

Figure 9: The evolution of layer sparsity during the LT extracting process, visualised in a number of selected layers of the VGG16 network.

In Appendix C, we showed that some layers are disproportionally pruned less by different pruning criteria. In this section, we will take a look at how the sparsity of these layers evolves throughout the pruning process and compare them to 'normal' layers in both VGG16 (see Figure 9) and ResNet-18 (see Figure 10).

It can be clearly seen that generally the deeper the layer is within the model, the faster this layer is pruned. This has the effect that often the higher layers are 'ahead' of the global sparsity, while the lower layers are 'behind' of the global sparsity. The main exceptions here being the 1x1 convolutional layers, however we can still notice that the later 1x1 convolutions are pruned faster than the 1x1 convolutions.

Additionally, we can notice some differences in behaviour of the different pruning criteria, namely that softmax has a spiky evolution, meaning that at some timesteps, no connections are pruned in a layer, while at other timesteps large amounts of connections are pruned. The other criteria follow a much smoother approach, where at least every timestep some connections are removed from each layer, with Min-Max often following the global sparsity best.

