# OpenReview forum: "Considering Layerwise Importance in the Lottery Ticket Hypothesis"
_ICLR.cc/2023/Conference — Submitted to ICLR 2023_

### Official Review · Reviewer_NGvs · 2022-10-20

**Confidence:** 5
**Correctness:** 3
**Technical Novelty And Significance:** 2
**Empirical Novelty And Significance:** 1
**Recommendation:** 3

**Clarity, Quality, Novelty And Reproducibility:**

As said in the previous section, the paper is well written and easy to follow. The experiments are well described and the authors provide all the details to reproduce them. The novelty of the proposed method seems quite limited (more comments above).

**Strength And Weaknesses:**

The paper is well written and easy to follow. However, I think that the novel contribution is limited (a similar idea of layerwise importance of the weights is already present in Tanaka et al. 2020). In addition, the results of the experiments do not show a significant gain using the proposed method and I haven't found the comparison of different tickets particularly insightful.

**Summary Of The Paper:**

The authors propose to modify the LTH procedure by using a score that measures the layerwise importance in order to select the weights to be pruned. The authors propose a few possible scores and evaluate their performance on a set of experiments. The experiments show that in most of the cases the proposed method does not improve with respect to the standard LTH that uses magnitude in order to select the weights to be pruned. The authors also perform a comparison of lottery tickets obtained with the different scores showing that hey have a noticeable amount of overlapping connections and such connections seems to be more stable.

**Summary Of The Review:**

Even though the paper is well written and it provides a solid experimental evaluation of the proposed method, I believe that the novel theoretical contribution is very limited and the proposed method does not exhibit a significant gain with respect to the state of the art.

---

> ### Author Response · Authors · 2022-11-10
> **RE: Official Review of Paper1758 by Reviewer NGvs**
>
> Dear Reviewer NGvs,
>
> Thanks for your feedback.
> To the best of our knowledge, no mention has been made in [Tanaka20] of layerwise importances for weights. The only references we found in this category are [Lee2021] or https://openreview.net/forum?id=B1xgQkrYwS. We will continue to refine and expand our work and hope for a positive review in the future.
>
> We thank the reviewer for their feedback.
>
> The authors
>
> References:
> [Lee21] : Lee et al. 2021 “Layer-adaptive Sparsity for the Magnitude-based Pruning” ICLR’21
> [Tanaka20] : Tanaka et al. 2020 “Pruning neural networks without any data by iteratively conserving synaptic flow” NeurIPS’20

---

### Official Review · Reviewer_8dwk · 2022-10-21

**Confidence:** 3
**Correctness:** 3
**Technical Novelty And Significance:** 2
**Empirical Novelty And Significance:** 3
**Recommendation:** 3

**Clarity, Quality, Novelty And Reproducibility:**

Clarity/quality/reproducibility are good.

I feel that the paper somewhat lacks in novelty. I make this statement because:
* Layer collapse and the issues of global IMP for LTH have already been observed
* Layer-wise pruning techniques have also been proposed and explored in depth
* The main proposed techniques for layer-wise LTH are L1/L2 normalization (minmax/softmax perform worse in nearly all cases), which is believe is fair to say is straightforward.

Despite my opinion on novelty, I believe novelty does exist in the authors' analysis of different winning tickets achieved with their methods. The novelty could be improved by 1) considering/deriving a wider set of potential layer-based importance metrics or 2) emphasizing and extending the analysis of the properties/overlap of tickets achieved via these different layer based techniques and normal IMP (e.g., can we use these different tickets to produce better ensembles or something similar?)

**Strength And Weaknesses:**

Strength:
* The LTH problem considered is popular and the focus on an intuitive/simple issue with typical approaches to LTH (i.e., global pruning leads and layer collapse). I believe this is an interesting direction that the authors motivate well.
* I believe the observation that many different structures of lottery tickets can be discovered within a network has already been made (see e.g., https://arxiv.org/abs/1912.05671), but the authors do provide some extra analysis that goes beyond previous work that is interesting/useful. I really enjoy this extra analysis, and I think the authors could emphasize this more!

Weakness:
* Because the paper is purely empirical, I believe that large-scale experiments on ImageNet are very necessary in this case. This is because, for the LTH, it is widely recognized that many observations made on LTH for smaller-scale experiments do not hold at scale (see https://arxiv.org/abs/1810.05270 and https://arxiv.org/abs/1902.09574). Thus, in this work, performing the larger-scale experiments (e.g., ImageNet) is necessary.
* Accuracy improvements of the proposed procedure are small and there is not a single procedure that performs best. The proposed methods are outperformed by magnitude pruning (i.e., the normal LTH procedure) in many of the considered cases.
* Several pruning works consider pruning in a layer-normalized fashion (see https://proceedings.mlsys.org/paper/2020/hash/d2ddea18f00665ce8623e36bd4e3c7c5-Abstract.html and references, just search “layer wise vs. global pruning” in the paper). None of these techniques are compared against within the analysis. I believe comparing against a few of the SOTA layer-based pruning techniques would be useful.

Small Stuff:
* Figure 1 is difficult to parse. It might be better to use a different subplot for each layer, then plot the sparsity as a function of the pruning iteration.
* L1/L2 normalization of per-layer importance are basically identical (even in the results). It might be better to just use one or the other and state that they perform very similarly.
* In the introduction, your posing of LTH makes it seem like it solves the problem of the train-prune-finetune loop. It might be good to state that LTH still requires this loop (i.e., the network needs to be pre-trained to discover the winning ticket still).
* “Using importance metrics makes this bias both more explicit and stronger, dependent on which metric is used.”: I am not quite sure what this means here or why this would be the case. It might be nice to explain a bit more or maybe provide a citation.
* “The only computational overhead incurred in our method is the calculation of importance scores, which is negligible.”: Whether this computational cost is negligible depends on the importance score you choose/implement, right?

Questions:
* Is there any motivation for the importance metrics that you consider, or are you simply trying a bunch of different approaches and trying to determine the best strategy empirically? Are there other approaches beyond these defined importances that could be useful?

**Summary Of The Paper:**

This paper considers the popular lottery ticket hypothesis (LTH) framework, which discovers sparse, trainable subnetworks via iterative magnitude pruning (IMP). Typically, IMP is performing in a global manner, in which weight magnitudes are compared with all other weights within a network when determining which weights should be pruned/discarded. Alternatively, the authors argue that comparing weights on a per-layer basis is more appropriate, as importances/magnitudes of weights oftentimes differ on a layer-to-layer basis throughout the network. Instead, they propose an alternative procedure that evaluates the importance of weights within each layer. In particular, this method (1) adopts the same weight magnitude and (2) normalizes this magnitude in a per-layer fashion using a few different techniques. This technique is evaluated for 4 different types of normalization on computer vision experiments. The authors then go on to show that different layer-based pruning techniques can be used to find winning tickets with different structures.

**Summary Of The Review:**

To begin, I want to emphasize that my current review reflects my initial impression of the paper. I am fully open to discussion with authors/other reviewers, and my final score will be mostly determined by this discussion.

I believe the problem the authors study and the proposed approach are interesting. However, I think the paper--in its current form--falls short of providing enough valuable insight to the problem of improving upon global IMP for LTH. Currently, the main weaknesses of this paper in my opinion are:
* Lack of large-scale experiments. This is very necessary for LTH as mentioned above.
* Lack of novelty in layer-wise importance schemes. The work could be greatly improved by exploring more schemes (possibly inspired by work in the pruning literature mentioned above).
* Improvements over normal LTH procedures are not consistent. I am not convinced the layer-wise importance approach truly provides a clear benefit.

Nonetheless, the paper is well-written and motivated, and I strongly encourage the authors to continue working in solving the problems I outline above. I especially enjoy the analysis of lottery ticket overlap at the end of the paper. I believe by furthering extending this analysis, thinking more deeply about other layer-wise techniques that could be used/compared-to, and expanding the empirical scope to more domains (e.g., ImageNet, transformer tasks, or other larger-scale cases) the authors will greatly improve the paper and provide a valuable contribution to the community.

---

> ### Author Response · Authors · 2022-11-10
> **RE: Official Review of Paper1758 by Reviewer 8dwk**
>
> Dear reviewer 8dwk,
>
> First and foremost, we thank you for your extensive and constructive feedback.
>
> **Regarding the size of the dataset.** We do concur that LTH findings on smaller datasets do not necessarily scale up to larger datasets as noticed in the linked papers [Liu19, Gale19]. However, we should stress that rather than the LTH with resetting [Frankle19] which is considered in the linked papers, we instead use the LTH procedure with rewinding (as introduced in [Frankle20]). In that paper, the authors demonstrate that, with rewinding, LTs do scale up to larger settings. While ImageNet is too computationally heavy for our computing budget, we did include the other settings that were considered in [Frankle21], which are similar to those of [Frankle20]. If TinyImageNet is considered not extensive enough, we can make an effort to upscale to a subset of the ImageNet dataset (e.g. containing the same 200 classes as in TinyImageNet), but we should stress that due to the nature of the LTH this will incur a hefty computational cost, namely (#random_seeds x #pruning_iterations x #considered_importances) full training runs for each network structure considered.
>
> **Regarding layerwise vs global pruning.** The reference of [Blalock20] studies a number of different layerwise & global methods, however in their case the layerwise methods prune exactly the same amount of connections in each layer. This means that for n layers, p/n connections are pruned in each layer. In the case of global pruning, p connections are pruned over the network, but not necessarily uniformly distributed over each layer. In our paper, we study an in-between. First we rescale the weights in each layer, with the explicit goal to make the importances of weights comparable between layers. Then we globally prune based on those importances. The most comparable techniques we found were either [Lee2021] or https://openreview.net/forum?id=B1xgQkrYwS. We will revise our paper to ensure this difference is more clear. Having said that, we do however acknowledge the idea of the reviewer that these importance methods do not constitute a lot of technical novelty per se nor result in incremental improvement, but we are happy that the reviewer sees the merit of the subsequent analysis we conducted.
>
> **Expanding the research.** As mentioned above, larger scale tasks are currently not feasible for us, as such the expansion of our work will mostly focus on refining the analysis in the second part of the paper. One provable improvement of Layerwise importance is that it can prevent Layer Collapse from occurring, for which we have a proof. In the first draft of our paper, this was located in the appendix, however in the final [submitted] draft we removed it from the appendix as we felt that it detracted too much from the analysis. If found valuable we could bring it back to the paper.
>
> We thank you for your review and hope to have a fruitful discussion,
>
> The authors.
>
> References:
>
> [Blalock20] : Blalock et al. 2020 “What is the State of Neural Network Pruning” MLSys’20
>
> [Frankle19] : Frankle & Carbin 2019 “The Lottery Ticket Hypothesis: Finding Sparse, Trainable Neural Networks” ICLR’19
>
> [Frankle20] : Frankle et al. 2020 “Linear Mode Connectivity and the Lottery Ticket Hypothesis” ICLR’20
>
> [Frankle21] : Frankle et al. 2021 “Pruning Neural Networks at Initialization: Why Are We Missing the Mark?” ICLR’21
>
> [Gale19] : Gale et al. 2019 “The State of Sparsity in Deep Neural Networks” ArXiv’19
>
> [Lee21] : Lee et al. 2021 “Layer-adaptive Sparsity for the Magnitude-based Pruning” ICLR’21
>
> [Liu19] : Liu et al. 2019 “Rethinking the Value of Network Pruning” ICLR’19

---

> > ### Comment · Reviewer_8dwk · 2022-11-15
> > **Response to authors**
> >
> > 1.
> > * I completely understand that running training on ImageNet is computationally difficult.
> > * Given that the paper is purely empirical, running training on ImageNet is (in my opinion) not an unreasonable request. ImageNet training can be run in a few days on a single GPU. Plus, for pruning experiments, one can simply use pre-trained models, prune them, then run fine-tuning, which is in general less expensive than running full training.
> > * In general, I regret that this places somewhat of a barrier to entry on deep learning, given that one needs access to some GPUs to perform empirical work.
> > * But, I cannot confidently say that this approach is useful without comparison to baselines on the most commonly-used large-scale dataset. Indeed, prior work has shown that LTH can generalize to large scale experiments. But, this says nothing about whether the proposed approach will. Without proof that the layerwise techniques shown in your work apply at scale, I cannot properly compare them to prior work.
> > * Obviously, the claims above seemingly leave no avenue to move forward unless the authors get more GPUs (which oftentimes is not possible). In this situation, I would recommend that the authors heavily emphasize the aspects of their work that do not require more GPUs. Namely, focus on extending the analysis as much as possible. For inspiration/direction, the original LTH paper is a good place to start -- they only run smaller-scale experiments, but the analysis is extremely extensive. What more can you analyze within winning tickets that is not present in prior work? You could also explore options like providing theory for layerwise pruning (many works exist along this line).
> >
> > 2. Understood. It would be good to compare/contrast your work with all of this research in writing. Providing a discussion of how your work relates to these different techniques is important. Additionally, including these techniques as baselines within your experiments where appropriate is extremely important.
> >
> > 3. I think that providing this analysis would be useful. Because you cannot run large scale experiments, you need to do your best to make contributions from other directions (i.e., theory and analysis). I believe including this result and extra analysis is the right direction for the paper.
> >
> > Overall, addressing these concerns is likely too large to accomplish in a revision. So, I advise the authors to continue working and resubmit to a later conference -- I think the paper currently does not meet standards for ICLR.
> >
> > With this being said, I believe the work can be made useful if the authors continue to pursue it. My recommendations above (i.e., focusing on more analysis and theory) will hopefully serve as a starting point. Being able to run large-scale experiments might not be possible. However, in this case, the authors need to compensate by providing a valuable contribution in other aspects of the work. Basically, this paper should be framed as an extensive analysis of different kinds of layerwise pruning techniques, how they work, and properties of the resulting tickets. The current analysis is a good start, and continuing in this direction will be fruitful in my opinion. Here are a few related works (I am not an author in any of these papers, simply providing them for reference) to provide some inspiration for different directions of analysis:
> > * https://arxiv.org/abs/2107.07467
> > * https://arxiv.org/abs/1912.05671
> > * https://arxiv.org/abs/1906.02773
> > * https://arxiv.org/abs/2010.03533
> > * https://arxiv.org/abs/1905.01067
> > * https://arxiv.org/pdf/1909.11957.pdf
> >
> > I thank the authors for their efforts and wish them the best of luck in further developing this work.

---

### Official Review · Reviewer_GSCn · 2022-10-24

**Confidence:** 3
**Clarity, Quality, Novelty And Reproducibility:** None
**Correctness:** 3
**Technical Novelty And Significance:** 1
**Empirical Novelty And Significance:** 2
**Recommendation:** 3

**Strength And Weaknesses:**

### Strength:

1. This paper is well-written and easy to follow.
2. Both quantitative and qualitative results are reported.
3. Experiments are conducted across different datasets and architectures.

### Weakness:

1. Only small-scale datasets are considered in the experiments.
2. I'm a little bit confused about the main contribution of this work. Using a layerwise importance score can improve the performance of identified lottery tickets or some observations that suggest lottery tickets are not unique. With different mini-batch sequences during training, we may also obtain significantly different lottery tickets from the same initialization.
3. Using layerwise importance score, can the LTH procedure identifies the winning ticket with higher sparsity?

**Summary Of The Paper:**

This work proposes a new Lottery Tickets procedure by introducing a layerwise importance score to find the lottery tickets. Additionally, this work demonstrates several intriguing observations: 1) from a fixed initialization, there are different lottery tickets that significantly differ in their structure. 2) these tickets have some common connections, which survive the LTH procedure even when the other weights are reinitialized.

**Summary Of The Review:**

None

---

> ### Author Response · Authors · 2022-11-10
> **RE: Official Review of Paper1758 by Reviewer GSCn**
>
> Dear reviewer GSCn,
>
> Thank you very much for the feedback.
> The main contributions of our work are: (1) a comprehensive comparison between different importance measures, notably showing that it is possible to find lottery tickets with some of these measures; (2) a comparative analysis of these resulting lottery tickets including the observation that there is a lot of overlapping connections between these tickets (when tickets are trained following the same fixed training procedure); and (3) the observation that these overlapping tickets are important for winning tickets, via conducting a partial randomization test.
> We will improve the writing of our paper in order to ensure this is more clear.
>
> Regarding the large-scale experiments, as also noticed in our response to reviewer 8dwk, it is computationally infeasible to study large-scale datasets – more specifically ImageNet – given our computational budget. We would appreciate it if you could have a look at the response we provided therein.
>
> Kind regards,
> The Authors

---

### Decision · Program_Chairs · 2023-01-20

**Decision:**

Reject

**Justification For Why Not Higher Score:**


Given the lack of a significant novel contribution, limited experiments on small-scale datasets, and absence of significant gains compared to the state of the art, I believe that this paper is not ready for publication at this stage.

**Justification For Why Not Lower Score:**

N/A

**Metareview: Summary, Strengths And Weaknesses:**


The Lottery Ticket Hypothesis (LTH) proposes that it is possible to extract small subnetworks at initialization that can be trained to full accuracy, through algorithms that typically involve pruning connections with the lowest global weight magnitude. However, as the authors of this paper argue this approach may not consider the context of the connections within a layer, as the weight distributions in different layers can vary. In this paper, the authors explore ways to recover this context by considering the importance of connections based on the weights of other connections within the same layer, rather than simply their absolute magnitude. They also generalize the LTH to use these importance metrics rather than weight magnitudes. Through experimentation on various architectures and datasets, the authors found that applying different importance metrics can result in distinct, high-performing "lottery tickets" with little overlap in connections.

The reviewers found that the work had a number of strengths. They noted that the research had both quantitative and qualitative results, and conducted experiments on a variety of datasets and architectures. Furthermore, they commented that the work was well written and easy to follow.

However, the reviewers identified several weaknesses in the work. Primarily, the lack of a novel contribution, as well as the experiments being limited to small-scale datasets. Furthermore, the proposed method does not show a significant gain compared to the state of the art LTH algorithms. Lastly, it is uncertain whether using a layerwise importance score can significantly help to identify sparser "winning tickets".

Overall, some reviewers engaged post the rebuttal session, however, the feedback provided by the authors was not convincing enough (in terms of timeline of implementation) to lift some of these criticisms. In particular, reviewer 8dwk provided extensive and constructive feedback to the authors, suggesting that they should focus more on analysis and theory in their paper and include comparison and contrast with related works. The reviewer also suggested that the authors should run experiments on a larger scale dataset, such as ImageNet, if possible. The authors responded to the feedback by explaining the technical differences between their work and the related works, and their plans to expand the research by refining the analysis and potentially bringing back a proof regarding Layerwise importance to the paper. However, (and I am in agreement with this) the changes requested, although possible, go beyond the scope of this round of reviews.

The authors are encouraged to take into the account all the feedback provided, and perhaps resubmit to a future venue.


**Summary Of Ac-Reviewer Meeting:**

N/A